**Data Availability Statement:** The minimal data set relevant to this study is included in the Supporting information. All data in Supporting information can be used without restriction. As for raw data, CMS

# Effect of common maintenance drugs on the risk and severity of COVID-19 in elderly patients

**Kin Wah Fung** ⬤*, **Seo H. Baik, Fitsum Baye, Zhaonian Zheng, Vojtech Huser, Clement J. McDonald**

Lister Hill National Center for Biomedical Communications, National Library of Medicine, National Institutes of Health, Bethesda, Maryland, United States of America

\* kfung@mail.nih.gov

## Abstract

### Background

Maintenance drugs are used to treat chronic conditions. Several classes of maintenance drugs have attracted attention because of their potential to affect susceptibility to and severity of COVID-19.

### Methods

Using claims data on 20% random sample of Part D Medicare enrollees from April to December 2020, we identified patients diagnosed with COVID-19. Using a nested case-control design, non-COVID-19 controls were identified by 1:5 matching on age, race, sex, dual-eligibility status, and geographical region. We identified usage of angiotensin-converting enzyme inhibitors (ACEI), angiotensin-receptor blockers (ARB), statins, warfarin, direct factor Xa inhibitors, P2Y12 inhibitors, famotidine and hydroxychloroquine based on Medicare prescription claims data. Using extended Cox regression models with time-varying propensity score adjustment we examined the independent effect of each study drug on contracting COVID-19. For severity of COVID-19, we performed extended Cox regressions on all COVID-19 patients, using COVID-19-related hospitalization and all-cause mortality as outcomes. Covariates included gender, age, race, geographic region, low-income indicator, and co-morbidities. To compensate for indication bias related to the use of hydroxychloroquine for the prophylaxis or treatment of COVID-19, we censored patients who only started on hydroxychloroquine in 2020.

### Results

Up to December 2020, our sample contained 374,229 Medicare patients over 65 who were diagnosed with COVID-19. Among the COVID-19 patients, 278,912 (74.6%) were on at least one study drug. The three most common study drugs among COVID-19 patients were statins 187,374 (50.1%), ACEI 97,843 (26.2%) and ARB 83,290 (22.3%). For all three outcomes (diagnosis, hospitalization and death), current users of ACEI, ARB,

did not allow the authors to download or distribute any patient level data. The data stayed in their machine and the authors analyzed it with software they provide on their machine. The shared detailed statistical data should be sufficient for anyone to verify the study's results. If researchers wish to access the raw data, they can contact the CMS Virtual Research Data Center. However, data access requires the payment of a fee. Note that the exact set of subjects may not be available even with access to raw data, because this study is based on a randomly selected 20% sample, and CMS will pull a new 20% sample for any new request. However, the results should be almost identical given the large sample size.

**Funding:** This research was supported in part by the Intramural Research Program of the NIH, National Library of Medicine. The funders had no role in study design, data collection and analysis, decision to publish, or preparation of the manuscript.

**Competing interests:** The authors have declared that no competing interests exist.

statins, warfarin, direct factor Xa inhibitors and P2Y12 inhibitors were associated with reduced risks, compared to never users. Famotidine did not show consistent significant effects. Hydroxychloroquine did not show significant effects after censoring of recent starters.

## Conclusion

Maintenance use of ACEI, ARB, warfarin, statins, direct factor Xa inhibitors and P2Y12 inhibitors was associated with reduction in risk of acquiring COVID-19 and dying from it.

## 1. Background and significance

Maintenance drugs are indicated for chronic conditions, taken indefinitely on regular, usually daily basis. Reports have attributed protective or aggravating effects of some maintenance drugs on the likelihood and/or severity of SARS-CoV-2 infection. These include angiotensin-converting enzyme inhibitors (ACEIs), angiotensin-receptor blockers (ARBs), statins, antithrombotic agents, hydroxychloroquine, and famotidine.

ACEI and ARB attracted early attention because they increase the expression of angiotensin-converting enzyme 2 (ACE2) [1, 2], the cellular doorway for SARS-CoV-2 [3, 4]. There are concerns that their use could *increase* the likelihood, or the severity, of a COVID-19 infection. However, most studies published to-date show no such increases [5–9], and some actually suggest that they *might decrease* COVID-19 severity [10, 11]. Due to the potential beneficial effect, several ongoing prospective trials are testing whether initiating ACEI or ARB treatment for patients after they are diagnosed with COVID-19 would confer any benefits [12–14].

Statins are among the commonest maintenance drugs in the elderly population. Attention was drawn to statins because of their potential effect on virus entry, replication or degradation, and their well-known anti-inflammatory properties [15, 16]. Several subsequent studies have shown the beneficial effects of statins in COVID-19 patients [17, 18].

Given the thrombotic propensity of COVID-19 [19–23], some think that antithrombotics could reduce the severity of COVID-19, and accordingly, anticoagulants are now recommended for hospitalized and high risk patients [24], though results of studies of maintenance antithrombotics have been mixed [25–29]. Famotidine has attracted attention because it was shown to have potential anti- SARS-CoV-2 activity based on structural homology modeling [30], and it improved COVID-19 outcomes in small studies [31–33]. Hydroxychloroquine was considered as treatment option for COVID-19 because of its immune modulation and anti-SARS-CoV-2 activity in vitro [34–36], and reported benefit in one clinical study [37].

Most of the early primary studies on the effect of these maintenance drugs have been small (tens or hundreds of COVID-19 patients who were also on the drugs) and would most likely lack statistical power to see important associations.

In the U.S., the Virtual Research Data Center (VRDC) [38] of the Centers for Medicare and Medicaid Services (CMS) carries de-identified data which include complete drug prescription, diagnoses, encounters as well as geographic, socio-demographic and vital status information for most (>93%) of the 65 and older U.S. residents. Over 80% of COVID-19 deaths occur in this age group, according to the U.S. Centers for Disease Control and Prevention (CDC). Therefore, this would be a propitious population in which to explore the association of maintenance drugs with COVID-19 risks.

In this study, we used the VRDC data and extended Cox regression analysis to examine associations between the above eight drug types and the occurrence of three outcomes: 1) COVID-19 infection, 2) COVID-19 hospitalization, and 3) death after a COVID19 diagnosis.

## 2. Materials and methods

### 2.1 Study population and case definition

VRDC provided us with a 20% random sample of all Medicare Part D enrollees. This set included 374,299 patients 65 or above, who had at least one record of the COVID-19 specific ICD10-CM diagnosis code of U07.1 between April 1 and December 31, 2020. We used both inpatient and outpatient claims and Medicare's vital status to identify COVID-19 cases and the outcome events. We only counted COVID-19 cases occurring on or after April 1, 2020, when the specific code for COVID-19 first became available. We stopped accruing cases after December 31, 2020, because of potential incompleteness of data due to the time lag (at least 3–4 months) between data capture and availability through VRDC. This study was declared not human subject research by the Office of Human Research Protection at the National Institutes of Health and by the CMS's Privacy Board.

### 2.2 Drugs, exposure definition and comorbidities

We identified all study drug preparations that were available in the U.S. market. For drug classes we identified the class members through Anatomical Therapeutic Chemical (ATC) classification codes and used generic drug names to find them in the CMS data. Our eight study drugs/drug classes were ACEI, ARB, statins, warfarin, direct factor Xa inhibitors, P2Y12 inhibitors, famotidine, and hydroxychloroquine (see S1 Table for full list of drugs and prescription frequencies). H2 blockers other than famotidine were not included because the potential beneficial effect was based on the chemical structure of famotidine and not the pharmacological class. Note that hydroxychloroquine was a class which also included two other aminoquinolines (chloroquine and primaquine), but hydroxychloroquine accounted for 99.7% of prescriptions.

We assumed that patients were on a given study drug during the window from the prescription dispensing date to 30 days after the end-of-supply day (we called this period the "current use period"). We added the 30-day buffer after the end-of-supply day because of the common behavior of drug stockpiling—patients maintaining a stock of drugs so that they will not run out immediately in case refill is interrupted. Drug use status was treated as a time-varying covariate in the Cox regression model. We defined a patient to be a current user if an outcome event fell within a current use period, former user if the event fell outside the current use period, and never user if they never had a prescription for that drug. Our primary analysis compared current users with never users as control. In a supplementary analysis, we used former users as control to see if that would give different results.

In the first few months of the pandemic, there were media reports of the potential beneficial or harmful effects of some of the study drugs on COVID-19. Publicity about the good effects of hydroxychloroquine was particularly rife, which could have led people to start taking it because of symptoms, or fear, of COVID -19. We looked at the trend of the usage of the study drugs, starting from January 2019, to identify abnormal patterns that could be related to COVID-19. To study the potential effect of COVID-19 affecting the use of drugs ("reverse causality"), we did separate analyses with special treatment of patients who were only recently started on a study drug (see section 2.3 below).

Medicare specifies the onset of 67 chronic conditions by algorithm and we followed the algorithm to define the occurrence and onset date of each condition [39]. To adjust for illness

burden in our analysis we included 57 Medicare chronic conditions with >1% prevalence in the Master Beneficiary Summary File.

### 2.3 Statistical analysis and covariates

We excluded patients who were not enrolled in Parts A (hospital) and B (medical) to ensure complete capture of hospitalization and encounter diagnoses. We considered the effect of the study drugs on three outcomes: 1) the risk of acquiring a COVID-19 diagnosis, 2) the risk of COVID-19 hospitalization, and 3) the risk of death after being diagnosed with COVID-19. For the first outcome (diagnosis of COVID-19), we used a nested case-control analysis [40], where each index COVID-19 patient was matched to patients with no COVID-19 diagnosis up to the date COVID-19 was diagnosed in the index patient.

For the first outcome (COVID-19 diagnosis), each index case was matched to five controls on age in years, race, sex, dual-eligibility status and five regions of residence (Northeast, Midwest, South, West, and others) at the time of the diagnosis of COVID-19. All cases and control were followed from January 1, 2020, until COVID-19 diagnosis, death, disenrollment from Medicare Parts A/B/D or December 31, 2020, whichever came first.

For the analyses of the second and third outcomes (hospitalization and death), we included only patients diagnosed with COVID-19. For the hospitalization outcome, we followed all patients from COVID-19 diagnosis until COVID-19-related hospitalization, or the other censoring points (except COVID-19 diagnosis) as described for the first outcome. For the death outcome, we did the same, swapping hospitalization for death.

We approximated patient's income level using the monthly indicators of dual-eligibility and low-income subsidy (LIS), which divided patients into three income groups: 1) dual-eligible: income ≤ 135% federal poverty line (FPL), 2) non-dual LIS: income > 135% and ≤ 150% FPL, and 3) non-dual no LIS: income > 150% FPL. To explore the effect of each study drug, we employed extended Cox regression analysis with days-on-study as time scale. We included age, gender, race, regions of residence, Medicare insurance type (Advantage or fee-for-service) and the degree of LIS as covariates. We also included binary flags for each of the 57 Medicare chronic conditions to control for the effect of co-morbidities. In order to protect against the immortal time bias and violation of proportional hazard assumption [41], we treated the use of each of the study drugs and almost all covariates as time-varying covariates. Only age, sex, race, and regions of residence were time-fixed, and their values were defined as of their values on January 1, 2020. Since the disease covariates were chronic diseases, we considered them as always present after their onset and always absent before then. The values of all time-varying covariates were reset at the time of each event in the Cox regression. We used Efron's adjustment for tied events.

To mitigate the potential of COVID-19 influencing the use of drugs, we did a separate Cox regression in which patients who only started a study drug in 2020 (they had never been on the drug before January 1,2020) were censored when they first started the drug ("recent starter censoring"). We compared the results with and without such censoring.

In order to mitigate selection bias toward use of the study drugs, we developed time-varying propensity scores (PS) [42] separately for each study drug using logistic regressions. The PS was the likelihood of receiving a study drug, conditional on patient's characteristics (demographics, socioeconomics, and presence of the chronic conditions). We iteratively estimated the PSs every month among the patients who remained in follow-up, considering all covariates that preceded the end of a given monthly cycle [43], and ran all Cox regression analyses with time-varying PSs as additional adjustments.

## 3. Results

### 3.1 Study population

In our 20% random sample of Part D enrollees, 374,299 Medicare beneficiaries aged 65 and above were diagnosed with COVID-19 between April 1 and December 31, 2020, among which 65,108 (17.4%) died (Table 1 and S1 Fig). Over the same period, CDC registered a national total of 19,852,636 COVID-19 cases and 350,510 deaths [44]. In the CDC statistics, even though patients ≥65 accounted for only 13.9% of cases, they accounted for 80.5% of mortalities. Therefore, our study population represents patients most at risk for COVID-19 death. Projecting from the CDC statistics, a 20% random sample of patients 65 and above would have 551,903 COVID-19 cases and 56,432 mortalities (Table 1). Therefore, our study population captured 67.8% of COVID-19 cases nationally. Our mortality number is 15% higher than the projected national count. Our death numbers were higher because Medicare data did not distinguish between causes of death. It is possible that some mortalities were not related to COVID-19. However, most of the mortalities occurred within a short time after the COVID-19 diagnosis (median 14 days, inter-quartile range 6–36 days). Compared with national statistics, the Medicare population was over-represented in females, whites and under-represented in Hispanics. Geographically, there was over-representation of the North-East region.

### 3.2 Drug exposure and matching

Table 2 shows the breakdown of drug usage (all patients who had prescription for the drug during the follow up period, starting on January 1, 2020) among the COVID-19 patients and controls. Overall, 278,912 (74.6%) cases and 1,351,244 (72.4%) controls had prescriptions for at least one study drug. The three most common study drugs among COVID-19 patients were statins 187,374 (50.1%), ACEI 97,872 843 (26.2%) and ARB 83,290 (22.3%). For all study drugs except ACEI and ARB, the usage was significantly higher among COVID-19 patients than controls. The unmatched rate (less than five controls found) was 0.4%.

**Table 1. Study population compared to U.S. national statistics, April to December 2020 (SRS—Simple random sample; CDC—Centers for Disease Control and Prevention).**

| | Medicare 20% SRS Age 65+ population (Apr-Dec 2020) | | National statistics from CDC (Apr-Dec 2020) | |
|---|---|---|---|---|
| | *Covid-19 cases (%)* | *Mortality (%)* | *Covid-19 cases* | *Mortality* |
| Total | 374,299(100) | 65,108(100) | 551,903* | 56,432* |
| Female | 225,585(60.3) | 36,093(55.0) | 52.2% | 45.7% |
| *Race* | | | | |
| White | 284,707(76.1) | 46,595(71.6) | 50.0% | 58.6% |
| Black | 38,419(10.3) | 8,702(13.4) | 11.0% | 13.6% |
| Hispanic | 31,981(8.5) | 6,401(9.8) | 29.2% | 18.9% |
| Asian | 9,961(2.7) | 2,032(3.1) | 3.3% | 4.0% |
| Other | 9,231(2.5) | 1,378(2.1) | 6.6% | 5.0% |
| *Geographic region* | | | | |
| Northeast | 82,281(22.0) | 16,292(25.0) | 17.15% | 23.45% |
| Midwest | 90,921(24.3) | 15,449(23.7) | 21.76% | 20.18% |
| South | 139,567(37.3) | 24,248(37.2) | 38.95% | 36.91% |
| West | 60,830(16.3) | 9,046(13.9) | 21.71% | 19.03% |

*The national case and mortality counts are extrapolations from the CDC data. They only include patients aged 65+, and reduced to 20% to be comparable with our 20% random sample.

**Table 2. Characteristics of Covid-19 patients and matched controls.**

| | Covid-19 patients (%) | Control (%) | (Control -Covid) Difference (95%CI) | PVALUE |
|---|---|---|---|---|
| **Drug exposure** | | | | |
| ACE Inhibitor | 97,843(26.2) | 517,078(27.7) | 1.5(1.4,1.7) | 0.000 |
| ARB | 83,290(22.3) | 421,264(22.6) | 0.3(0.2,0.4) | 0.000 |
| Statin | 187,374(50.1) | 915,226(49.0) | -1.1(-1.2,-0.9) | 0.000 |
| Warfarin | 11,755(3.1) | 47,251(2.5) | -0.6(-0.7,-0.6) | 0.000 |
| Direct Factor Xa Inhibitor | 42,599(11.4) | 161,365(8.6) | -2.7(-2.9,-2.6) | 0.000 |
| P2Y12 Inhibitor | 40,199(10.7) | 157,173(8.4) | -2.3(-2.4,-2.2) | 0.000 |
| Hydroxychloroquine | 2,879(0.8) | 11,846(0.6) | -0.1(-0.2,-0.1) | 0.000 |
| Famotidine | 13,133(3.5) | 40,984(2.2) | -1.3(-1.4,-1.3) | 0.000 |
| Any drug | 278,912(74.6) | 1,351,244(72.4) | -2.2(-2.3,-2.0) | 0.000 |
| No Rx | 95,094(25.4) | 515,332(27.6) | 2.2(2.0,2.3) | 0.000 |
| **Age** | | | | |
| 65–69 | 37,859(10.1) | 189,150(10.1) | 0.0(-0.1,0.1) | 0.839 |
| 70–74 | 89,538(23.9) | 447,650(24.0) | 0.0(-0.1,0.2) | 0.582 |
| 75–79 | 77,520(20.7) | 387,477(20.8) | 0.0(-0.1,0.2) | 0.662 |
| 80–84 | 64,951(17.4) | 324,637(17.4) | 0.0(-0.1,0.2) | 0.704 |
| 85 + | 104,138(27.8) | 517,662(27.7) | -0.1(-0.3,0.0) | 0.168 |
| **Region** | | | | |
| MIDWEST | 90,840(24.3) | 453,053(24.3) | -0.0(-0.2,0.1) | 0.830 |
| NORTHEAST | 82,225(22.0) | 410,408(22.0) | 0.0(-0.1,0.1) | 0.976 |
| SOUTH | 139,505(37.3) | 696,758(37.3) | 0.0(-0.1,0.2) | 0.747 |
| WEST | 60,761(16.2) | 303,161(16.2) | -0.0(-0.1,0.1) | 0.946 |
| OTHER | 675(0.2) | 3,196(0.2) | -0.0(-0.0,0.0) | 0.213 |
| Unmatched | 293 ($<$ 0.1) | - | | |
| **Total** | **374,299 (100)** | **1,866,576 (100)** | | |
| Number of patients with $<$ 5 matches | 1446 (0.4) | | | |

As for abnormal patterns of drug usage related to COVID-19, we detected a sharp rise in the use of hydroxychloroquine in March and April 2020 (S2 Fig), which was not seen in other study drugs. FDA granted emergency use authorization for hydroxychloroquine for COVID-19 treatment on March 28, and the percentage of COVID-19 patients on hydroxychloroquine surged to over 0.8% compared to the baseline of 0.3% before the pandemic. There was a similar but smaller rise in non-COVID-19 patients, showing that some patients might be taking hydroxychloroquine for prophylaxis or suspicion of COVID-19. After FDA revoked the emergency use authorization on June 15, the usage of hydroxychloroquine began to drop, but remained slightly above the baseline before the pandemic.

### 3.3 Risk of being diagnosed with COVID-19

The total number of patients followed (cases and controls) was about 2.2 million, and around 350,000 ended up with the diagnosis of COVID-19 (Table 3). We did separate analyses with and without recent starter censoring for all study drugs. Only hydroxychloroquine exhibited significantly different results with censoring. We report the results with recent starter censoring for hydroxychloroquine as the main results in Table 3. The results for hydroxychloroquine without such censoring are shown for comparison.

Compared to patients who never used the drug, current users of ACEI, ARB, statin, warfarin, direct factor Xa inhibitors and P2Y12 inhibitors were associated with decreased risk of

**Table 3. Effect of drug use on Covid-19 diagnosis, Covid-19 hospitalization and death (other non-drug covariates are listed in Table 4).**

| | Covid-19 diagnosis | | Covid-19 hospitalization | | Death | |
|---|---|---|---|---|---|---|
| | No. of patients (events) | Hazard ratio (95% CI) | No. of patients (events) | Hazard ratio (95% CI) | No. of patients (events) | Hazard ratio (95% CI) |
| **Drug use status: current vs. never** | | | | | | |
| ACE Inhibitor | 2,185,934 (354,342) | **0.91(0.90,0.92)** | 358,392 (142,004) | **0.98(0.97,1.00)** | 358,392 (61,778) | **0.88(0.86,0.90)** |
| ARB | 2,185,934 (354,342) | **0.92(0.91,0.92)** | 358,392 (142,004) | **0.92(0.91,0.94)** | 358,392 (61,778) | **0.85(0.83,0.87)** |
| Statin | 2,185,934 (354,342) | **0.97(0.96,0.98)** | 358,392 (142,004) | **0.95(0.94,0.96)** | 358,392 (61,778) | **0.81(0.80,0.83)** |
| Warfarin | 2,185,934 (354,342) | **0.88(0.86,0.91)** | 358,392 (142,004) | **0.95(0.92,0.99)** | 358,392 (61,778) | **0.82(0.78,0.87)** |
| Direct Factor Xa Inhibitor | 2,185,934 (354,342) | **0.99(0.97,1.00)** | 358,392 (142,004) | **0.89(0.88,0.91)** | 358,392 (61,778) | **0.80(0.78,0.82)** |
| P2Y12 Inhibitor | 2,185,934 (354,342) | **0.98(0.97,0.99)** | 358,392 (142,004) | **0.96(0.95,0.98)** | 358,392 (61,778) | **0.94(0.91,0.96)** |
| Famotidine | 2,185,934 (354,342) | *1.12(1.10,1.15)* | 358,392 (142,004) | **0.94(0.91,0.97)** | 358,392 (61,778) | 1.00(0.96,1.04) |
| Hydroxychloroquine (New Users Censored) | 2,185,934 (354,342) | 0.95(0.91,1.00) | 358,392 (142,004) | 1.06(0.98,1.14) | 358,392 (61,778) | 1.08(0.95,1.24) |
| Hydroxychloroquine (No Censoring) * | 2,186,365 (357,499) | *1.63(1.58,1.68)* | 361,568 (143,728) | 0.97(0.93,1.01) | 361,568 (62,698) | 1.06(0.99,1.14) |
| **Drug use status: current vs. past** | | | | | | |
| ACE Inhibitor | 2,185,934 (354,342) | **0.86(0.85,0.87)** | 358,392 (142,004) | **0.95(0.93,0.97)** | 358,392 (61,778) | **0.87(0.84,0.90)** |
| ARB | 2,185,934 (354,342) | **0.93(0.91,0.94)** | 358,392 (142,004) | **0.93(0.91,0.96)** | 358,392 (61,778) | **0.84(0.82,0.87)** |
| Statin | 2,185,934 (354,342) | **0.92(0.91,0.93)** | 358,392 (142,004) | **0.93(0.92,0.95)** | 358,392 (61,778) | **0.79(0.77,0.81)** |
| Warfarin | 2,185,934 (354,342) | **0.84(0.81,0.87)** | 358,392 (142,004) | **0.91(0.86,0.95)** | 358,392 (61,778) | **0.83(0.77,0.90)** |
| Direct Factor Xa Inhibitor | 2,185,934 (354,342) | **0.89(0.87,0.91)** | 358,392 (142,004) | **0.88(0.85,0.90)** | 358,392 (61,778) | **0.79(0.76,0.82)** |
| P2Y12 | 2,185,934 (354,342) | 1.01(0.99,1.03) | 358,392 (142,004) | 0.99(0.96,1.02) | 358,392 (61,778) | **0.90(0.87,0.94)** |
| Famotidine | 2,185,934 (354,342) | **0.93(0.91,0.96)** | 358,392 (142,004) | **0.95(0.92,0.99)** | 358,392 (61,778) | 1.03(0.97,1.09) |
| Hydroxychloroquine (New Users Censored) | 2,185,934 (354,342) | **0.91(0.84,0.98)** | 358,392 (142,004) | 0.97(0.86,1.09) | 358,392 (61,778) | 0.87(0.72,1.05) |
| Hydroxychloroquine (No Censoring) * | 2,186,365 (357,499) | *1.35(1.28,1.42)* | 361,568 (143,728) | 1.12(1.03,1.22) | 361,568 (62,698) | *1.27(1.14,1.42)* |

**Bold**: significant protective effect; *Italic*: significant harmful effect.

*not part of main analysis, shown here to illustrate the indication bias if recent starters of hydroxychloroquine were not censored.

contracting COVID-19, ranging from 1% reduction (direct factor Xa inhibitors) to 12% reduction (warfarin). For famotidine, there was a 12% increase in risk. For hydroxychloroquine, with the proper censoring of recent starters, there was no effect on the risk of being diagnosed with COVID-19. However, without censoring, current users of hydroxychloroquine would appear to be associated with an increased risk of getting COVID-19 compared to never users (63% increase).

The results for the non-drug covariates are shown in Table 4. Compared with the youngest group of patients in our population (65–69), older age was associated with decreased risk of COVID-19 diagnosis. Female sex was associated with a 11% risk reduction compared to male. The risk of COVID-19 diagnosis was higher in all race groups compared to whites, 13% higher for Asians, 54% for Blacks and 85% for Hispanics. The comorbidities associated with the greatest increase in risk were dementia (132% increase), hypertension (125% increase) and schizophrenia (68% increase).

### 3.4 Risk of COVID-19 hospitalization

Overall, 142,004 (39.6%) COVID-19 patients were hospitalized for COVID-19. Current users of ACEI, ARB, statin, warfarin, direct factor Xa inhibitors, P2Y12 inhibitors and famotidine were associated with 2–11% decreased risk of COVID-19 hospitalization when compared with

**Table 4. Effect of non-drug covariates on Covid-19 diagnosis, Covid-19 hospitalization and death (drug-related covariates are listed in Table 3).**

| Co-variate | Reference | COVID-19 diagnosis | COVID hospitalization | Death |
|---|---|---|---|---|
| | | Hazard ratio (95% CI) | | |
| **Demographics** | | | | |
| 70–74 | 65–69 | **0.98(0.97,0.99)** | 1.10(1.08,1.12) | 1.12(1.07,1.16) |
| 75–79 | 65–69 | **0.83(0.81,0.84)** | 1.40(1.36,1.43) | 1.55(1.49,1.61) |
| 80–84 | 65–69 | **0.75(0.74,0.76)** | 1.63(1.59,1.66) | 2.03(1.95,2.12) |
| >85 | 65–69 | **0.83(0.81,0.85)** | 1.95(1.90,2.01) | 3.33(3.19,3.48) |
| Female | Male | **0.89(0.87,0.90)** | **0.80(0.79,0.82)** | **0.65(0.63,0.68)** |
| Black | white | 1.54(1.52,1.57) | 0.99(0.97,1.01) | 1.04(1.01,1.07) |
| Hispanic | white | 1.85(1.82,1.88) | 1.19(1.17,1.22) | 1.31(1.27,1.35) |
| Asian | white | 1.13(1.08,1.17) | 0.99(0.94,1.04) | 1.26(1.18,1.35) |
| Other | white | 1.09(1.07,1.12) | 1.07(1.04,1.11) | 1.32(1.25,1.40) |
| Dual | Non-dual No LIS | **0.45(0.44,0.45)** | 1.11(1.08,1.13) | **0.64(0.62,0.66)** |
| Non-dual LIS | Non-dual No LIS | **0.82(0.80,0.83)** | 1.47(1.42,1.51) | 3.06(2.95,3.17) |
| Midwest | Northeast | 1.15(1.14,1.17) | 1.43(1.40,1.46) | 1.08(1.04,1.11) |
| South | Northeast | 1.00(0.99,1.01) | 1.25(1.23,1.27) | 1.00(0.97,1.02) |
| West | Northeast | 1.67(1.65,1.70) | 1.23(1.20,1.26) | 0.97(0.94,1.00) |
| Other | Northeast | **0.83(0.76,0.91)** | **0.41(0.33,0.52)** | **0.54(0.42,0.71)** |
| **Comorbidities** | | | | |
| ESRD | No | **0.86(0.83,0.90)** | **0.75(0.72,0.78)** | 1.28(1.21,1.36) |
| AMI | No | **0.70(0.69,0.71)** | 1.17(1.14,1.20) | 1.39(1.35,1.44) |
| Atrial Fibrillation | No | **0.30(0.30,0.31)** | **0.92(0.89,0.95)** | 1.24(1.18,1.30) |
| Cataract | No | 1.08(1.07,1.09) | **0.84(0.83,0.85)** | **0.80(0.78,0.82)** |
| Chronic Kidney Disease | No | 0.99(0.98,1.00) | 1.34(1.32,1.35) | 1.56(1.53,1.60) |
| COPD | No | **0.97(0.96,0.98)** | **0.95(0.94,0.96)** | 1.04(1.02,1.07) |
| Heart Failure | No | **0.96(0.95,0.96)** | 1.01(1.00,1.02) | 1.13(1.10,1.15) |
| Diabetes | No | 1.06(1.05,1.08) | 1.18(1.16,1.19) | 0.98(0.96,1.01) |
| Glaucoma | No | **0.97(0.96,0.97)** | 1.00(0.99,1.02) | **0.97(0.95,0.98)** |
| Hip/Pelvic Fracture | No | 1.47(1.45,1.50) | 1.11(1.09,1.13) | 1.16(1.13,1.19) |
| Ischemic Heart Disease | No | **0.70(0.69,0.71)** | **0.90(0.88,0.92)** | **0.87(0.84,0.89)** |
| Depression | No | 1.04(1.03,1.05) | **0.96(0.94,0.97)** | **0.96(0.94,0.98)** |
| Alzheimer'S Disease Or Dementia | No | 2.32(2.28,2.35) | 1.32(1.29,1.34) | 2.27(2.21,2.34) |
| Osteoporosis | No | **0.85(0.84,0.86)** | **0.81(0.80,0.82)** | **0.85(0.84,0.87)** |
| Rheumatoid Arthritis/Osteoarthritis | No | 0.99(0.98,1.01) | **0.84(0.83,0.86)** | **0.81(0.79,0.83)** |
| Stroke/Transient Ischemic Attack | No | **0.76(0.75,0.77)** | 1.00(0.98,1.01) | 1.00(0.97,1.02) |
| Breast Cancer | No | 1.01(0.99,1.02) | 0.99(0.97,1.02) | 0.97(0.94,1.01) |
| Colorectal Cancer | No | 1.03(1.01,1.04) | **0.95(0.93,0.98)** | 1.03(0.99,1.06) |
| Prostate Cancer | No | 1.01(1.00,1.02) | **0.96(0.94,0.99)** | 0.99(0.96,1.02) |
| Lung Cancer | No | 0.98(0.96,1.00) | 0.98(0.95,1.02) | 1.53(1.46,1.59) |
| Endometrial Cancer | No | 1.10(1.06,1.13) | 1.06(1.01,1.10) | 1.06(0.99,1.13) |
| Anemia | No | 1.07(1.06,1.09) | **0.94(0.93,0.96)** | 1.05(1.02,1.07) |
| Asthma | No | **0.83(0.82,0.84)** | **0.87(0.86,0.88)** | **0.81(0.79,0.83)** |
| Hyperlipidemia | No | **0.57(0.55,0.59)** | **0.79(0.76,0.83)** | **0.58(0.54,0.62)** |
| Hyperplasia | No | **0.96(0.95,0.97)** | **0.87(0.85,0.89)** | **0.90(0.88,0.93)** |
| Hypertension | No | 2.25(2.17,2.33) | 2.34(2.21,2.47) | 0.95(0.88,1.03) |
| Hypothyroidism | No | 1.00(0.99,1.01) | **0.87(0.86,0.88)** | **0.91(0.90,0.93)** |
| ADHD And Other Conduct Disorders | No | 1.24(1.21,1.26) | 0.99(0.96,1.01) | 1.13(1.09,1.17) |
| Alcohol Use Disorders | No | 1.14(1.12,1.16) | 1.03(1.01,1.05) | 1.00(0.97,1.03) |

(*Continued*)

**Table 4.** (Continued)

| Co-variate | Reference | COVID-19 diagnosis | COVID hospitalization | Death |
|---|---|---|---|---|
| | | Hazard ratio (95% CI) | | |
| Anxiety Disorders | No | *1.11(1.10,1.12)* | **0.98(0.97,1.00)** | *1.07(1.05,1.09)* |
| Bipolar Disorder | No | *1.13(1.11,1.14)* | *1.08(1.07,1.10)* | *1.05(1.03,1.08)* |
| Traumatic Brain Injury | No | *1.19(1.17,1.22)* | *1.07(1.04,1.10)* | 1.04(1.00,1.08) |
| Drug Use Disorder | No | **0.91(0.90,0.93)** | *1.05(1.02,1.07)* | 0.98(0.94,1.01) |
| Intellectual Disabilities | No | **0.79(0.77,0.81)** | **0.88(0.85,0.91)** | **0.91(0.86,0.96)** |
| Learning Disabilities | No | *1.08(1.05,1.12)* | 1.00(0.96,1.04) | *1.06(1.00,1.13)* |
| Other Developmental Delays | No | **0.73(0.69,0.76)** | **0.84(0.79,0.90)** | 0.91(0.82,1.01) |
| Personality Disorders | No | *1.03(1.01,1.04)* | *1.04(1.02,1.07)* | 0.98(0.95,1.01) |
| Schizophrenia | No | *1.68(1.66,1.70)* | *1.04(1.02,1.05)* | *1.19(1.16,1.22)* |
| Post-Traumatic Stress Disorder | No | **0.85(0.82,0.87)** | **0.93(0.89,0.96)** | **0.89(0.83,0.95)** |
| Cerebral Palsy | No | *1.22(1.17,1.26)* | **0.90(0.85,0.95)** | 0.92(0.84,1.01) |
| Epilepsy | No | *1.04(1.03,1.05)* | **0.93(0.91,0.95)** | 0.99(0.96,1.01) |
| Cystic Fibrosis | No | *1.11(1.09,1.12)* | 0.98(0.96,1.00) | *1.04(1.01,1.07)* |
| Fibromyalgia, Chronic Pain And Fatigue | No | **0.93(0.92,0.93)** | **0.89(0.88,0.90)** | **0.83(0.81,0.84)** |
| Viral Hepatitis (General) | No | *1.23(1.20,1.26)* | *1.04(1.01,1.07)* | *1.09(1.04,1.14)* |
| Liver Disease Cirrhosis | No | **0.99(0.98,1.00)** | **0.95(0.94,0.96)** | *1.03(1.01,1.05)* |
| Hiv/Aids | No | *1.10(1.05,1.15)* | **0.91(0.85,0.98)** | 0.92(0.83,1.02) |
| Leukemias And Lymphomas | No | **0.98(0.96,0.99)** | 1.03(1.00,1.06) | *1.24(1.19,1.29)* |
| Migraine And Other Chronic Headache | No | **0.78(0.77,0.79)** | **0.80(0.79,0.82)** | **0.74(0.71,0.76)** |
| Mobility Impairments | No | **0.90(0.89,0.91)** | **0.97(0.95,0.98)** | **0.97(0.95,1.00)** |
| Multiple Sclerosis And Transverse Myelitis | No | *1.11(1.07,1.14)* | *1.15(1.10,1.20)* | 1.05(0.98,1.13) |
| Obesity | No | **0.96(0.95,0.97)** | *1.16(1.14,1.18)* | **0.90(0.88,0.92)** |
| Overarching Opioid Use Disorder | No | **0.93(0.91,0.94)** | **0.95(0.92,0.97)** | **0.94(0.90,0.98)** |
| Peripheral Vascular Disease | No | *1.08(1.07,1.09)* | *1.05(1.03,1.06)* | *1.05(1.03,1.08)* |
| Spinal Cord Injury | No | *1.14(1.12,1.16)* | *1.07(1.05,1.10)* | *1.09(1.06,1.13)* |
| Tobacco Use Disorders | No | **0.98(0.96,0.99)** | *1.07(1.05,1.09)* | *1.08(1.05,1.11)* |
| Pressure Ulcers And Chronic Ulcers | No | *1.22(1.20,1.23)* | **0.98(0.96,0.99)** | *1.41(1.38,1.44)* |
| Deafness And Hearing Impairment | No | **0.92(0.91,0.93)** | **0.94(0.93,0.95)** | **0.95(0.93,0.97)** |
| Blindness And Visual Impairment | No | *1.27(1.25,1.30)* | *1.03(1.01,1.06)* | *1.12(1.08,1.17)* |

**Bold**: significant protective effect; *Italic*: significant harmful effect.

never users (Table 3). Hydroxychloroquine did not show any effect when recent starters were censored. The risk of COVID-19 hospitalization increased monotonically with age (Table 4). Compared to the 65–69 age group, the risk increased by 10%, 40%, 63% and 95% for groups 70–74, 75–79, 80–84 and ≥85 respectively. Female sex was associated with a 20% reduced risk compared to male. The risk of COVID-19 hospitalization for Hispanics was 19% higher than whites. Poorer patients were associated with higher risks of hospitalization with COVID-19. The three co-morbidities associated with largest increase in risk were hypertension (134%), chronic kidney disease (34%) and dementia (32%).

## 3.5 Risk of mortality

Overall, 61,778 (17.2%) COVID-19 patients died. Current users of ACEI, ARB, statin, warfarin, direct Xa inhibitors and P2Y12 inhibitors among COVID-19 patients were all associated with lower risks of mortality, from 6% less than never users (P2Y12 inhibitors) to 20% less

(direct factor Xa inhibitors) (Table 3). Famotidine and hydroxychloroquine showed no significant effect. Mortality risk increased monotonically with age (Table 4). Compared to patients 65–69, all other age groups exhibited increased risk of death: 12%, 55%, 103% and 233% increased risk among 70–74, 75–79, 80–84 and ≥85 respectively. Female sex was associated with a 35% reduced risk of death compared to male. All race groups were associated with notably greater mortality risk than whites. Unlike analyses of other outcomes, the poorest patients (dual-eligible) were associated with 36% decreased risk of death, while the second poorest patients (non-dual LIS) had 206% increased risk of death compared to patients not dually eligible and not receiving LIS. The three co-morbidities associated with the greatest mortality risk were dementia (127%) chronic kidney disease (56%) and lung cancer (53%).

## 4. Discussion

The most important finding of our study is that after controlling for age, gender, race, socioeconomic, geographic factors and co-morbidities, there was an associated decline in mortality risk of 12% or greater among current users for ACEIs, ARB, statins, and anticoagulants, and by a lesser amount (6%) among P2Y12 inhibitors users. To the best of our knowledge, ours is first single source study to show such beneficial associations, though it has been hinted at in meta-analyses [10, 11]. While more study is needed to identify the exact reasons for severity and mortality modification of the study drugs in COVID-19 patients, some possible mechanisms have been proposed. One predominant hypothesis is that SARS-CoV-2 down-regulates ACE2 expression, resulting in unabated angiotensin II activity that may be responsible for organ damage in COVID-19 [6]. ACEI and ARB reduce angiotensin II activity and so are protective [45]. Our findings provide strong support to the advice from professional societies and the WHO that ACEI and ARB be continued in COVID-19 patients [46, 47]. The benefits of antithrombotics can be explained by reduction in the thromboembolism observed in COVID-19 patients [21, 23, 48]. These results strongly support the current advice about the use of anti-clotting drugs in COVID-19 patients. The benefits of statins can be associated with their anti-viral and anti-inflammatory properties [15, 16]. In addition, our study gives justification for the need of clinical trials to initiate treatment with drugs that are potentially beneficial (such as ACEI, ARB and statins) in patients diagnosed with COVID-19.

The main strength of this study is the size of the study population and relative completeness of demographic, prescription, and co-morbidity data. The U.S. has the largest collection of COVID-19 cases in the world. The elderly represents the most at-risk population for severe COVID-19. Our study population covered over 370,000 COVID-19 cases and 65,000 mortalities in U.S. patients over 65. According to CDC statistics, over 80% of COVID-19 mortalities occurred in patients over 65. A large U.K. study based on NHS records registered 10,926 COVID-19 deaths, our study surpassed that number by almost six-fold [49]. In terms of study drug usage among COVID-19 patients, our numbers were also much higher. For instance, one of the biggest early original ACEI studies was from Italy, covering 1,502 ACEI users among COVID-19 patients [7]. We had 65 times that number. Recently published studies do have larger patient numbers. One such study supported by the American Heart Association's Rapid Response Grant COVID-19 is by An et al. [50], with 4,652 patients on ACEI and 2,546 on ARB. However, this is still lower than the number of patients observed in our study, 97,843 on ACEI and 83,290 on ARB. Sample sizes of this magnitude are not seen in other studies, including meta-analyses [8, 10, 11]. Moreover, other studies tended to be restricted to institutional or regional populations. Our Medicare population includes most US elderly individuals. By one estimate 93% of all US adults over 65 are enrolled to Medicare [51]. Unlike meta-analysis that pools multiple, not necessarily comparable data sources, we used a single data source that has

relatively complete and longitudinal medication information with well-documented data capture procedures.

The emergence of hydroxychloroquine as a potential "game-changer" in COVID-19 had been accompanied by the dramatic rise in its use, not only among COVID-19 patients but in the general population as well. Even after the emergency use authorization was revoked by FDA, the use of hydroxychloroquine remained somewhat higher than the baseline level before the pandemic. This shows that media claims, even those that are eventually shown to be unsubstantiated or disproved, may have lasting impact on the public psyche. In our observational study, the use of hydroxychloroquine is a classic example of indication bias—the indication for the exposure is directly related to the outcome being observed [52]. Since we have longitudinal prescription data (starting from 2019), we were able to compensate for this by censoring patients who were first started on hydroxychloroquine during the pandemic. Indeed, our results show that, without censoring, the use of hydroxychloroquine would have appeared to be associated with an increase in the risk of COVID-19 diagnosis and death, but the effects disappeared with proper censoring.

Regarding the risk of catching COVID-19, we found that ACEI, ARB, statins and antithrombotics were associated with a reduction in the risk of getting a COVID-19 diagnosis. One possible explanation is that if these drugs blunted the effects of COVID-19 infection, infected patients might have milder symptoms and would be more likely to not seek medical care and remain undiagnosed. The apparent reduction in risk of catching COVID-19 with advancing age could be related to the reduced level of social activity at extreme age, thus lowering the risk of exposure.

Apart from drug usage, we found the following risk factors for severe COVID-19— advanced age, male, non-white, and co-morbidities including chronic kidney diseases, dementia, hypertension, heart diseases, chronic obstructive pulmonary disease, chronic liver diseases and some malignancies. These findings concur with similar studies and are well-documented. In our study, the group with the lowest income (dual-eligible) was associated with a reduced risk of death. One possible explanation is that this group has better access to healthcare, since patients are eligible to both Medicaid and Medicare. However, this explanation only applies to the Medicaid and Medicare patients and not low income patients in general.

We recognize the following limitations. In our retrospective observation study, drug exposure was implied from prescription records and there was no verification that the drugs were dispensed or taken. Drug data from Medicare claims are incomplete for hospitals and nursing homes. As a result, the continuation of medications for inpatients could not be ascertained. For certain drugs (e.g., statins), it has been reported that rebound effects were associated with drug discontinuation which could potentially affect the outcomes of COVID-19 infection [53]. Some of the study drugs are often prescribed in combination with other drugs. For example, statins are often used together with anti-hypertensive medications. In our Medicare population, statins are taken together with ACEI (in 35% of patients on statins), ARB (28.6%), beta blockers (44.3%), calcium channel blockers (37.7%) and diuretics (39.8%). (S2 Table) This can be a potential confounder if not properly addressed. In our multiple variable analysis, the use of ACEI, ARB and statins are included in the same unified regression model as time-varying covariates. This ensures that the observed effects of each individual study drug are already adjusted for the co-prescription of other drugs included in the model [54]. It is true that the combined use of statins with other anti-hypertensive drugs (e.g., diuretics, beta blockers) has not been adjusted for. However, up to the start of the study, there was no strong evidence that diuretics, calcium channel blockers or beta blockers significantly affect the susceptibility to and severity of COVID-19. It is therefore unlikely that the observed effects of the study drugs on COVID-19 are caused by these other drugs. Furthermore, there could also be other residual

confounding factors outside the scope of our regression models. We did not include COVID-19 patients before April because the COVID-19 specific diagnosis code was not in use. Compared with national data from CDC, we could be missing about one-third of COVID-19 cases. It is likely that milder cases would be missed since the patients did not seek medical care. This would also explain the relatively high hospitalization and mortality rate in our population compared to some other studies [55]. We used all-cause mortality because of the lack of an accurate cause of death in CMS data. Because of this, it is possible that some reduction of mortality is due to the general protective effect of the study drugs and not related to COVID-19 per se (e.g., statins and hypotensive agents can reduce cardiovascular mortality). However, since the mortality rate was 17% in our COVID-19 patients overall, 29% among those hospitalized, and death occurred within a median of 14 days of COVID-19 diagnosis, it is highly likely that COVID-19 was the main contributor to mortality. Another potential confounding factor is that some patients could be diagnosed with COVID-19 because of more frequent routine testing during an encounter for another reason (e.g., preparation for coronary bypass), which makes COVID-19 an incidental finding rather than the trigger event. However, overall, only 3.2% of our Medicare population were diagnosed with COVID-19, so the chance of a purely incidental COVID-19 diagnosis is small.

## 5. Conclusion

Analysis of more than 370,000 Medicare enrollees over 65 diagnosed with COVID-19 showed that the use of ACEI, ARB, statins, warfarin, direct factor Xa inhibitors and P2Y12 inhibitors was associated with a reduction in the risk of catching COVID-19 and developing severe disease. Hydroxychloroquine and famotidine were not associated with significant effects in these outcomes.

## Supporting information

**S1 Fig. Consort diagram.**
(PDF)

**S2 Fig. Trends of drug usage in 2019 and 2020.**
(PDF)

**S1 Table. Drug classes, clinical drugs and usage frequencies among all our study population.**
(DOCX)

**S2 Table. The combination use of anti-hypertensive drugs with statins in Medicare patients for 2019–2020.**
(DOCX)

**S3 Table. Data values for drug drug usage trend graphs in S2 Fig.**
(XLSX)

**S4 Table. Detailed statistical data of the Cox regression models for death, hospitalization and acquiring COVID-19.**
(XLSX)

## Acknowledgments

We would like to thank the diligent and helpful staff at the VRDC and Research Data Assistance Center (ResDAC), without them this study would not be possible.

## Author Contributions

**Conceptualization:** Kin Wah Fung, Seo H. Baik, Clement J. McDonald.

**Formal analysis:** Seo H. Baik, Fitsum Baye, Zhaonian Zheng.

**Investigation:** Vojtech Huser.

**Methodology:** Kin Wah Fung, Clement J. McDonald.

**Writing – original draft:** Kin Wah Fung, Seo H. Baik.

**Writing – review & editing:** Kin Wah Fung, Seo H. Baik, Fitsum Baye, Zhaonian Zheng, Vojtech Huser, Clement J. McDonald.

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
