## [Decision Letter · Decision Letter 0]

19 Oct 2021

PONE-D-21-31663Effect of common maintenance drugs on the risk and severity of COVID-19 in elderly patientsPLOS ONE

Dear Dr. Fung,

Thank you for submitting your manuscript to PLOS ONE. After careful consideration, we feel that it has merit but does not fully meet PLOS ONE’s publication criteria as it currently stands. Therefore, we invite you to submit a revised version of the manuscript that addresses the points raised during the review process.

Major revisions are needed in the present form. See the Reviewers' comments carefully and respond them appropriately.

We look forward to receiving your revised manuscript.

Kind regards,

Masaki Mogi

Academic Editor

PLOS ONE

Journal Requirements:

"No"

"No"

Reviewers' comments:

Reviewer's Responses to Questions

**Comments to the Author**

1. Is the manuscript technically sound, and do the data support the conclusions?

Reviewer #1: Partly

Reviewer #2: Yes

2. Has the statistical analysis been performed appropriately and rigorously? 

Reviewer #1: Yes

Reviewer #2: Yes

3. Have the authors made all data underlying the findings in their manuscript fully available?

Reviewer #1: No

Reviewer #2: Yes

4. Is the manuscript presented in an intelligible fashion and written in standard English?

Reviewer #1: Yes

Reviewer #2: Yes

5. Review Comments to the Author

Reviewer #1: This is nested case-control study derived from the Medicare claims database, to evaluate the association between various medications and the incidence and severity of COVID. In a 20% random sample of the database, COVID-19 cases were matched 1:5 with non-COVID controls. Analyses adjusted for various factors including comorbidities, geographical region, and insurance status. The authors accounted for indication bias for hydroxychloroquine, by censoring recent initiators of the medication. The authors found that several medications were associated with reduced risk of getting COVID and of severe COVID (hospitalization, death).

Overall this is a very interesting study and the authors have done a nice job presenting the findings. The very large dataset provides sufficient power to both evaluate and account for a large number of variables.

I have several comments:

Throughout the manuscript – but especially in the abstract and the Discussion, the authors need to be very careful to avoid implying causation from the observed associations.

Abstract Conclusion- “…was found to be protective against…” needs to be tempered to describe the association. i.e. “was associated with…”

Similarly, in the Discussion, instead of saying “risk declined by 15% or greater” – should say it was “associated with” a lower risk; and instead of “beneficial effects”, would say “beneficial associations”.

The choice of which medications to evaluate is not well explained. It does not make sense to me to include clopidogrel but not other agents of the same class. If including clopidogrel, you need to include the other P2Y12 agents: ticagrelor, prasugrel, (and ticlopidine – though less commonly used) – similar to how you look at ACEi’s and ARBs as classes. What is the rationale for only using clopidogrel?

Similarly, why only famotidine? Is the anti-viral activity unique to the medication and not the others within this class? If so, this should be stated and referenced, and you need to exclude subjects on the other H2 blockers. If not, should include all H2 blockers as a class.

Apixaban should be included as an anticoagulant as well.

In any observational study, some residual confounding may persist – and this limitation needs to be stated.

In addition, patients who are not on any prescription medications at all may differ from those on medications in important ways besides just health status – even though there are claims present to cause them to be included in the dataset, a subset may have less healthcare access, which could influence the associations seen. It would be of interest to see a sensitivity analysis excluding those who are not on any medications.

Methods -

The authors state “All cases and control were followed from January 1, 2020 until COVID-19 diagnosis, death, disenrollment from Medicare Parts A/B/D or December 31, 2020, whichever came first” – however cases must have been followed beyond their COVID diagnosis, or else outcomes such as death would not be known…Please specify that this is just the follow-up for the first outcome, risk of acquiring COVID.

Are you capturing all the COVID diagnoses? With a 40% hospitalization rate, it would seem that a lot of COVID cases are not actually being captured. Also, mild cases are probably often undiagnosed and thus not captured. You need a discussion of this in the Limitations section.

Table 3 and Table 4 are quite interesting – just a suggestions, but if color is allowed, it would be nice to use that instead of bold vs italic, to make the significant findings a bit easier to notice.

The Discussion section could benefit from a more thorough description and citing of prior literature.

Reviewer #2: This is an interesting and important paper. Its focus is on whether drugs that are often taken by older patients might reduce the occurrence of outcomes of SARS-CoV-2 infection. It is a direct examination of whether these drugs can be considered “repurposed” for COVID-19 care. They show that RAS inhibitors and several anticoagulant preparations can be helpful.

The strengths of the study are its very large size, its use of a single database and its very good statistical methods (time varying propensity scoring). It usefully shows that taking hydroxychloroquine was not helpful, a finding that should lay to rest any lingering hopes that it is. The study also shows that famotidine, a drug hyped as having potential benefits, was not beneficial. The study’s inevitable limitation is that it cannot ascertain how well its subjects were adhering to their prescribed medications in the days and weeks immediately before the diagnosis of SARS-CoV-2 infection was made.

The study has one gobsmackingly glaring omission. The authors have not included statins among the drugs they studied. Statins are taken considerably more frequently by older persons (>65 years of age) than any of the other drugs they studied. There are approximately 50 individual published reports of statin effects in COVID-19 patients and at least 14 meta-analyses of these reports. Twelve studies report inpatient statin treatment is always associated with reduced COVID-19 mortality.

Why did the authors not study statins? They must explain why they excluded statins from their study. If possible, they could re-examine their database to determine whether statin treatment was associated with any COVID-19 outcomes.

The authors must bear in mind that outpatient documentation of statin treatment is unable to document whether statin treatment was continued after hospital admission. There is a very real risk of a rebound effect following statin withdrawal (see Cubeddu LX et al. Pharmacotherapy 2006; 26:1288-96). Whether this effect follows inpatient withdrawal of any of the drugs included in the authors’ study is unknown; the evidence for withdrawal effects for discontinued ACE inhibitors and ARBs is mixed. However, this effect must be considered a possibility and must be discussed by the authors.

The authors should acknowledge that many of the drugs they studied are taken in combination preparations and the drugs with which they are combined might have their own effects on COVID-19 outcomes. For example, ARBs are often taken in combination with calcium channel blockers and CCBs by themselves are said to improve COVID-19 outcomes. A sensitivity analysis that focuses only on the effects of losartan (an ARB taken by 65% of all study subjects) and other ARBs alone might help settle this question.

6. PLOS authors have the option to publish the peer review history of their article (what does this mean?). If published, this will include your full peer review and any attached files.

Reviewer #1: No

Reviewer #2: No

---

## [Author Response · Author response to Decision Letter 0]

8 Dec 2021

PONE-D-21-31663

Effect of common maintenance drugs on the risk and severity of COVID-19 in elderly patients

Dear PLOS ONE editors,

The authors would like to thank the reviewers for their detailed review and thoughtful comments and suggestions. The following is our specific response to the comments.

Reviewer #1: This is nested case-control study derived from the Medicare claims database, to evaluate the association between various medications and the incidence and severity of COVID. In a 20% random sample of the database, COVID-19 cases were matched 1:5 with non-COVID controls. Analyses adjusted for various factors including comorbidities, geographical region, and insurance status. The authors accounted for indication bias for hydroxychloroquine, by censoring recent initiators of the medication. The authors found that several medications were associated with reduced risk of getting COVID and of severe COVID (hospitalization, death).

Overall this is a very interesting study and the authors have done a nice job presenting the findings. The very large dataset provides sufficient power to both evaluate and account for a large number of variables.

I have several comments:

Throughout the manuscript – but especially in the abstract and the Discussion, the authors need to be very careful to avoid implying causation from the observed associations.

Abstract Conclusion- “…was found to be protective against…” needs to be tempered to describe the association. i.e. “was associated with…”

Response: The verbiage throughout the manuscript has been changed as suggested.

Similarly, in the Discussion, instead of saying “risk declined by 15% or greater” – should say it was “associated with” a lower risk; and instead of “beneficial effects”, would say “beneficial associations”.

Response: The verbiage throughout the manuscript has been changed as suggested.

The choice of which medications to evaluate is not well explained. It does not make sense to me to include clopidogrel but not other agents of the same class. If including clopidogrel, you need to include the other P2Y12 agents: ticagrelor, prasugrel, (and ticlopidine – though less commonly used) – similar to how you look at ACEi’s and ARBs as classes. What is the rationale for only using clopidogrel?

Response: Thanks for the suggestion. We have modified the list of drugs as suggested. Clopidogrel is expanded to P2Y12 inhibitors, which include clopidogrel, prasugrel and ticagrelor, there are no patients on ticlopidine in our population.

Similarly, why only famotidine? Is the anti-viral activity unique to the medication and not the others within this class? If so, this should be stated and referenced, and you need to exclude subjects on the other H2 blockers. If not, should include all H2 blockers as a class.

Response: the suggested beneficial effect of famotidine is based on structural homology analysis [Wu et al] and does not apply to other H2 blockers. This has been clarified in the manuscript.

Apixaban should be included as an anticoagulant as well.

Response: thanks for the suggestion. Apixiban is now included in direct factor Xa inhibitors.

In any observational study, some residual confounding may persist – and this limitation needs to be stated.

Response: residual confounding factors is added as a limitation

In addition, patients who are not on any prescription medications at all may differ from those on medications in important ways besides just health status – even though there are claims present to cause them to be included in the dataset, a subset may have less healthcare access, which could influence the associations seen. It would be of interest to see a sensitivity analysis excluding those who are not on any medications.

Response: we have done the suggested analysis. Overall, among our COVID-19 patients and controls, only 2.5% did not have any prescription at all for the period 2019-2020. Because of the small percentage, we do not think that a separate subset analysis is warranted. 

Methods -

The authors state “All cases and control were followed from January 1, 2020 until COVID-19 diagnosis, death, disenrollment from Medicare Parts A/B/D or December 31, 2020, whichever came first” – however cases must have been followed beyond their COVID diagnosis, or else outcomes such as death would not be known…Please specify that this is just the follow-up for the first outcome, risk of acquiring COVID.

Response: clarification added to Methods

Are you capturing all the COVID diagnoses? With a 40% hospitalization rate, it would seem that a lot of COVID cases are not actually being captured. Also, mild cases are probably often undiagnosed and thus not captured. You need a discussion of this in the Limitations section.

Response: Compared with national data from CDC, we could be missing about one-third of COVID-19 cases. It is likely that milder cases would be missed since the patients did not seek medical care. This would also explain the higher COVID-19 hospitalization and mortality rate in our study compared to other studies. This is now added as a limitation

Table 3 and Table 4 are quite interesting – just a suggestions, but if color is allowed, it would be nice to use that instead of bold vs italic, to make the significant findings a bit easier to notice.

Response: results are color-coded as suggested

The Discussion section could benefit from a more thorough description and citing of prior literature.

Response: more references are added

Reviewer #2: This is an interesting and important paper. Its focus is on whether drugs that are often taken by older patients might reduce the occurrence of outcomes of SARS-CoV-2 infection. It is a direct examination of whether these drugs can be considered “repurposed” for COVID-19 care. They show that RAS inhibitors and several anticoagulant preparations can be helpful.

The strengths of the study are its very large size, its use of a single database and its very good statistical methods (time varying propensity scoring). It usefully shows that taking hydroxychloroquine was not helpful, a finding that should lay to rest any lingering hopes that it is. The study also shows that famotidine, a drug hyped as having potential benefits, was not beneficial. The study’s inevitable limitation is that it cannot ascertain how well its subjects were adhering to their prescribed medications in the days and weeks immediately before the diagnosis of SARS-CoV-2 infection was made.

Response: implying drug exposure from prescription records is added as limitation

The study has one gobsmackingly glaring omission. The authors have not included statins among the drugs they studied. Statins are taken considerably more frequently by older persons (>65 years of age) than any of the other drugs they studied. There are approximately 50 individual published reports of statin effects in COVID-19 patients and at least 14 meta-analyses of these reports. Twelve studies report inpatient statin treatment is always associated with reduced COVID-19 mortality.

Why did the authors not study statins? They must explain why they excluded statins from their study. If possible, they could re-examine their database to determine whether statin treatment was associated with any COVID-19 outcomes.

Response: Thanks for the suggestion. Statins have been added to the list of study drugs. Statins are indeed associated with beneficial effects. References are added regarding the effects of statins and the possible mechanisms.

The authors must bear in mind that outpatient documentation of statin treatment is unable to document whether statin treatment was continued after hospital admission. There is a very real risk of a rebound effect following statin withdrawal (see Cubeddu LX et al. Pharmacotherapy 2006; 26:1288-96). Whether this effect follows inpatient withdrawal of any of the drugs included in the authors’ study is unknown; the evidence for withdrawal effects for discontinued ACE inhibitors and ARBs is mixed. However, this effect must be considered a possibility and must be discussed by the authors.

Response: this is added as limitation, specifically mentioning the possible rebound effect mentioned by Cubeddu et al.

The authors should acknowledge that many of the drugs they studied are taken in combination preparations and the drugs with which they are combined might have their own effects on COVID-19 outcomes. For example, ARBs are often taken in combination with calcium channel blockers and CCBs by themselves are said to improve COVID-19 outcomes. A sensitivity analysis that focuses only on the effects of losartan (an ARB taken by 65% of all study subjects) and other ARBs alone might help settle this question.

Response: at the time of the study, we focused on the 8 maintenance drugs/drug classes that received the most attention in relation to COVID-19. There could be other drugs (e.g., calcium channel blockers) that could potentially affect the outcome of COVID-19 infection. But generally, the evidence associated with those drugs tends to be weaker and they were not included in our study. It is true that some of those drugs are frequently taken in combination with some of the study drugs. The possible confounding due to drug combination is added as a limitation.

---

## [Decision Letter · Decision Letter 1]

3 Jan 2022

PONE-D-21-31663R1Effect of common maintenance drugs on the risk and severity of COVID-19 in elderly patientsPLOS ONE

Dear Dr. Fung,

Thank you for submitting your manuscript to PLOS ONE. After careful consideration, we feel that it has merit but does not fully meet PLOS ONE’s publication criteria as it currently stands. Therefore, we invite you to submit a revised version of the manuscript that addresses the points raised during the review process.

Major revisions are still necessary in the present form of the manuscript.See the comments from the two Reviewers carefully and respond them appropriately.

We look forward to receiving your revised manuscript.

Kind regards,

Masaki Mogi

Academic Editor

PLOS ONE

Reviewers' comments:

Reviewer's Responses to Questions

**Comments to the Author**

1. If the authors have adequately addressed your comments raised in a previous round of review and you feel that this manuscript is now acceptable for publication, you may indicate that here to bypass the “Comments to the Author” section, enter your conflict of interest statement in the “Confidential to Editor” section, and submit your "Accept" recommendation.

Reviewer #1: (No Response)

Reviewer #2: All comments have been addressed

2. Is the manuscript technically sound, and do the data support the conclusions?

Reviewer #1: Partly

Reviewer #2: Yes

3. Has the statistical analysis been performed appropriately and rigorously? 

Reviewer #1: No

Reviewer #2: Yes

4. Have the authors made all data underlying the findings in their manuscript fully available?

Reviewer #1: Yes

Reviewer #2: Yes

5. Is the manuscript presented in an intelligible fashion and written in standard English?

Reviewer #1: Yes

Reviewer #2: Yes

6. Review Comments to the Author

Reviewer #1: The manuscript is improved, but some concerns remain:

An additional consideration is that very few individuals use a statin alone, in the absence of ACEI/ARBs (as well as anti-platelet medications). This number may be less than 7% of subjects. One must address the co-occurrence of these medications with care. Previous recent studies of medication use among patients with COVID-19 have shown that over 80% of patients on statins also take anti-hypertensive medication. These co-occurrences could be a substantial source of bias if not carefully addressed. I don’t see that this was addressed at all in statistical analyses or study design.

Intro

P14 – the authors state that almost all of the primary studies of these maintenance drugs are small. This is no longer true, especially for ACEi/ARB and statins. There are now studies with thousands and thousands of patients. (For instance, studies from the American Heart Association COVID-19 study teams, among others.)

Cause of death not specified – so unclear whether statins, ACE/ARB etc are reducing CVD mortality or COVID mortality. The fact that they often occurred a median of 14 days after a diagnosis could also reflect, in part, increased testing among any hospitalized patient during this time period (i.e. prior to any cardiac procedure or surgery.)

Limited to Medicare population – more access to healthcare, might not apply to other populations.

P21, last 2 paragraphs – grammar needs some attention (i.e. “older age”, not “older patients”, is associated with reduced risk; “current use of”, not “current uses of” hydroxychloroquine). Recommend “female sex” was “associated with” a reduced risk, not “Female experienced a 11% risk reduction” [sic]; and “older age”, not “older patients” were associated with decreased risk. Similar changes on p25 and p26 are recommended.

Tables 3 & 4 – Please list in the footnote for each Table what covariates these analyses are adjusted for.

The Discussion of why these classes of medications (especially ACEI/ARBS and statins) may be associated with benefits is grossly oversimplified.

.

Reviewer #2: Please see my comments to the editor and to the authors. I am uncertain about whether the authors have made their data available to others. See item 4 above.

7. PLOS authors have the option to publish the peer review history of their article (what does this mean?). If published, this will include your full peer review and any attached files.

Reviewer #1: No

Reviewer #2: No

---

## [Author Response · Author response to Decision Letter 1]

16 Feb 2022

PONE-D-21-31663

Effect of common maintenance drugs on the risk and severity of COVID-19 in elderly patients

Dear PLOS ONE editors,

The authors would like to thank the reviewers for their thoughtful comments and suggestions. The following is our specific response.

Reviewer #1: The manuscript is improved, but some concerns remain:

An additional consideration is that very few individuals use a statin alone, in the absence of ACEI/ARBs (as well as anti-platelet medications). This number may be less than 7% of subjects. One must address the co-occurrence of these medications with care. Previous recent studies of medication use among patients with COVID-19 have shown that over 80% of patients on statins also take anti-hypertensive medication. These co-occurrences could be a substantial source of bias if not carefully addressed. I don’t see that this was addressed at all in statistical analyses or study design.

Response: It is true that statins are often co-prescribed with other maintenance drugs, especially anti-hypertensives. In our Medicare population, statins are taken together with ACEI (in 35% of patients on statins), ARB (28.6%), beta blockers (44.3%) and diuretics (39.8%) (see newly added supplementary table 2). This can be a potential confounder if not properly addressed. In our multiple regression analysis, the use of ACEI, ARB and statins are included in the same unified regression model as time-varying covariates. This ensures that the observed effects of each individual study drug are already adjusted for the co-prescription of other drugs included in the model. A new reference [reference 54] is added to explain the statistical control of confounding effects. It is true that the combined use of statins with other anti-hypertensive (e.g., diuretics, beta blockers) has not been adjusted for. However, up to the start of the study, there is no strong evidence that diuretics, calcium channel blockers or beta blockers significantly affect the susceptibility to and severity of COVID-19. It is therefore unlikely that the observed effects of the study drugs on COVID-19 are caused by these other drugs. 

Intro

P14 – the authors state that almost all of the primary studies of these maintenance drugs are small. This is no longer true, especially for ACEi/ARB and statins. There are now studies with thousands and thousands of patients. (For instance, studies from the American Heart Association COVID-19 study teams, among others.)

Response: That sentence in the introduction has been modified to “Most of the early primary studies…have been small…”. In the Discussion, it is now stated that: “Recently published studies do have larger patient numbers. One such study supported by the American Heart Association’s Rapid Response Grant COVID-19 is by An et al. [new reference 50 added] with 4,652 patients on ACEI and 2,546 on ARB. However, this is still lower than the number of patients observed in our study, 97,843 on ACEI and 83,290 on ARB.”

Cause of death not specified – so unclear whether statins, ACE/ARB etc are reducing CVD mortality or COVID mortality. The fact that they often occurred a median of 14 days after a diagnosis could also reflect, in part, increased testing among any hospitalized patient during this time period (i.e. prior to any cardiac procedure or surgery.)

Response: This point is well taken. To elaborate on this issue, the following is added to Discussion: “We used all-cause mortality because of the lack of an accurate cause of death in CMS data. Because of this, it is possible that some reduction of mortality is due to the general protective effect of the study drugs and not related to COVID-19 per se (e.g., statins and hypotensive agents can reduce cardiovascular mortality). However, since the mortality rate was 17% in our COVID-19 patients overall, 29% among those hospitalized, and death occurred within a median of 14 days of COVID-19 diagnosis, it is highly likely that COVID-19 was the main contributor to mortality. Another potential confounding factor is that some patients could be diagnosed with COVID-19 because of more frequent routine testing during an encounter for another reason (e.g., preparation for coronary bypass), which makes COVID-19 an incidental finding rather than the trigger event. However, overall, only 3.2% of our Medicare population were diagnosed with COVID-19, so the chance of a purely incidental COVID-19 diagnosis is small.”

Limited to Medicare population – more access to healthcare, might not apply to other populations.

Response: The sentence is now qualified. “In our study, the group with the lowest income (dual-eligible) was associated with a reduced risk of death. One possible explanation is that this group has better access to healthcare, since patients are eligible to both Medicaid and Medicare. However, this explanation only applies to the Medicaid and Medicare patients and not low income patients in general.”

P21, last 2 paragraphs – grammar needs some attention (i.e. “older age”, not “older patients”, is associated with reduced risk; “current use of”, not “current uses of” hydroxychloroquine). Recommend “female sex” was “associated with” a reduced risk, not “Female experienced a 11% risk reduction” [sic]; and “older age”, not “older patients” were associated with decreased risk. Similar changes on p25 and p26 are recommended.

Response: text modified as suggested, thanks.

Tables 3 & 4 – Please list in the footnote for each Table what covariates these analyses are adjusted for.

Response: A note is added to the title of each table to refer to the other table for covariates that are adjusted for.

The Discussion of why these classes of medications (especially ACEI/ARBS and statins) may be associated with benefits is grossly oversimplified.

Response: Detailed discussion of the possible pharmacologic mechanisms for the benefits of the study drugs is beyond the scope of this paper. We can only highlight the most predominant hypotheses. Newer references [references 45, 48] are added for the possible pharmacologic effects of ACEI/ARB and antithrombotics. 

Reviewer #2: Please see my comments to the editor and to the authors. I am uncertain about whether the authors have made their data available to others. See item 4 above.

Response: We’ve submitted new data and modified the Data availability statement as follows: “Concerning data availability, the minimal data set is included in the Supporting information. All data in Supporting information can be used without restriction. This includes the precise values used to build the drug usage trend graphs (S3 Table), and the detailed statistical data obtained in the three Cox regressions (S4 Table), from which the hazard ratios can be derived. As for raw data, CMS does not let us download (or distribute) any patient level data. The data stay on their machine, and we analyze it with software they provide on their machine. The shared detailed statistical data, that we are allowed to take out of their machine, should be sufficient for anyone to verify our results. If researchers wish to access the raw data, they can contact the CMS Virtual Research Data Center. However, data access requires the payment of a fee. Note that the exact set of subjects may not be available even with access to raw data, because our study is based on a randomly selected 20% sample, and CMS will pull a new 20% sample for any new request. However, the results should be almost identical to ours, given the large sample size.”

---

## [Decision Letter · Decision Letter 2]

30 Mar 2022

Effect of common maintenance drugs on the risk and severity of COVID-19 in elderly patients

PONE-D-21-31663R2

Dear Dr. Fung,

We’re pleased to inform you that your manuscript has been judged scientifically suitable for publication and will be formally accepted for publication once it meets all outstanding technical requirements.

Kind regards,

Masaki Mogi

Academic Editor

PLOS ONE

Additional Editor Comments (optional):

Reviewers' comments:

Reviewer's Responses to Questions

**Comments to the Author**

1. If the authors have adequately addressed your comments raised in a previous round of review and you feel that this manuscript is now acceptable for publication, you may indicate that here to bypass the “Comments to the Author” section, enter your conflict of interest statement in the “Confidential to Editor” section, and submit your "Accept" recommendation.

Reviewer #1: All comments have been addressed

Reviewer #2: (No Response)

2. Is the manuscript technically sound, and do the data support the conclusions?

Reviewer #1: Yes

Reviewer #2: Yes

3. Has the statistical analysis been performed appropriately and rigorously? 

Reviewer #1: Yes

Reviewer #2: Yes

4. Have the authors made all data underlying the findings in their manuscript fully available?

Reviewer #1: Yes

Reviewer #2: Yes

5. Is the manuscript presented in an intelligible fashion and written in standard English?

Reviewer #1: Yes

Reviewer #2: Yes

6. Review Comments to the Author

Reviewer #1: I have no further comments for the authors; they have, overall, addressed the issues I have brought up.

Reviewer #2: The difference between the conflicting findings of outpatient documented statin treatment and uniform findings that inpatient treatment reduces COVID-19 severity and mortality is critically important. Documentation of statin treatment based only on out patient information does not take into account the effects of statin withdrawal after hospital admission. Moreover, if inpatients are treated with statins, treatment might be withdrawn if they are transferred to ICUs, although intravenously administered statins are licensed if not widely available.7 Whenever statins are withdrawn, their beneficial effects on the host response can be rapidly lost.8 For example, cardiovascular investigators who studied patients hospitalized with acute myocardial infarction 15–20 years ago found that those who had been treated with statins as outpatients and whose statins were continued after hospital admission had lower mortality rates than those who had never received statins.9 The same benefit was seen in those who were started on statin treatment after hospitals admission. However, those who had been treated with statins as outpatients but whose treatment was withdrawn after hospital is underway. In the absence of clinical trials, physicians may have to rely on the findings of observational studies alone.

7. PLOS authors have the option to publish the peer review history of their article (what does this mean?). If published, this will include your full peer review and any attached files.

Reviewer #1: No

Reviewer #2: No

---

## [Editor Report · Acceptance letter]

8 Apr 2022

PONE-D-21-31663R2 

Effect of common maintenance drugs on the risk and severity of COVID-19 in elderly patients 

Dear Dr. Fung:

I'm pleased to inform you that your manuscript has been deemed suitable for publication in PLOS ONE. Congratulations! Your manuscript is now with our production department. 

Kind regards, 

on behalf of

Dr. Masaki Mogi 

Academic Editor

PLOS ONE